# Adaptive Thermogenesis and Lipid Metabolism Modulation in Inguinal and Perirenal Adipose Tissues of *Hezuo Pigs* in Response to Low-Temperature Exposure

**DOI:** 10.3390/cells14060392

**Published:** 2025-03-07

**Authors:** Yao Li, Hai-Xia Shi, Jie Li, Hong Du, Rui Jia, Yu-Hao Liang, Xiao-Yu Huang, Xiao-Li Gao, Shuang-Bao Gun, Qiao-Li Yang

**Affiliations:** 1College of Animal Science and Technology, Gansu Agricultural University, Lanzhou 730070, China; ly926111@163.com (Y.L.); shxgsau@163.com (H.-X.S.); lijie5272@126.com (J.L.); duhong0905@163.com (H.D.); jiaruirui0920@163.com (R.J.); liangyuhao134134@163.com (Y.-H.L.); huanghxy100@163.com (X.-Y.H.); gxl18892@163.com (X.-L.G.); 2Gansu Modern Pig Rearing Engineering and Technology Research Center, Lanzhou 730070, China; 3Gansu Diebu Juema Pig Science and Technology Backyard, Gannan 740070, China

**Keywords:** *Hezuo pigs*, adipose tissue, cold exposure, lipid metabolism, RNA-seq

## Abstract

In mammals, exposure to low temperatures induces white adipose tissue (WAT) browning and alters lipid metabolism to promote thermogenesis, thereby maintaining body temperature. However, this response varies across different adipose depots. In this study, *Hezuo pigs* were exposed to either room temperature (23 ± 2 °C) or low temperature (−15 ± 2 °C) for periods of 12 h, 24 h, 48 h, 5 d, 10 d, and 15 d. Inguinal fat (IF) and perirenal fat (PF) were collected and analyzed using hematoxylin and eosin (HE) staining, transmission electron microscopy, RT-qPCR, and RNA-seq. Following cryoexposure, our results demonstrated a significant increase in adipocyte number and a corresponding decrease in cross-sectional area in both IF and PF groups from 24 h to 10 d. While adipocyte numbers were elevated at 12 h and 15 d, these changes were not statistically significant. Moreover, lipid droplets and mitochondria were more abundant, and the mRNA expression levels of thermogenic genes *UCP3* and *PGC-1α* were significantly higher compared to the control group during the 24 h-10 d cold exposure period. No significant changes were observed in the other groups. RNA-seq data indicated that the lipid metabolism of IF and PF peaked on day 5 of low-temperature treatment. In IF tissue, lipid metabolism is mainly regulated by genes such as *FABP4*, *WNT10B*, *PCK1*, *PLIN1*, *LEPR*, and *ADIPOQ*. These genes are involved in the classical lipid metabolism pathway and provide energy for cold adaptation. In contrast, in PF tissue, genes like *ATP5F1A*, *ATP5PO*, *SDHB*, *NDUFS8*, *SDHA*, and *COX5A* play roles within the neurodegenerative disease pathway, and PF tissue has a positive impact on the process related to degenerative diseases. Further investigation is needed to clarify the functions of these candidate genes in lipid metabolism in *Hezuo pigs* and to explore the genetic mechanisms underlying the cold-resistance traits in local pig populations.

## 1. Introduction

Cold is a major challenge for livestock and poultry farming in high-latitude regions, with profound implications for animal health, welfare, and productivity. Extensive research has demonstrated that prolonged cold exposure induces cold tolerance adaptations in animals, with notable interspecies and intraspecies variations. Among Chinese indigenous pig breeds, the *Hezuo pig* (Tibetan plateau type) exhibits exceptional cold adaptation, maintaining normal growth and reproductive functions at temperatures as low as −15 °C without thermal insulation. This remarkable cold resistance is particularly evident when compared to imported commercial breeds [1]. Investigating the physiological mechanisms underlying cold tolerance in *Hezuo pigs* is crucial for the effective utilization of their genetic potential and for the advancement of cold-resistant pig breeding programs.

In cold environments, animals usually adopt strategies to counteract the effects of low temperatures, with enhanced thermogenesis being a key one. Non-shivering thermogenesis, mainly driven by lipid metabolism in adipose tissue, is the primary mechanism for sustained cold adaptation. Adipose tissue plays a crucial role in cold resistance; the body adapts to low temperatures by regulating the balance between lipogenesis and lipolysis. Adipose tissue consists of white adipose tissue, mainly for lipid storage, and brown adipose tissue, specialized in energy expenditure [2]. Under certain stimuli, white adipocytes can undergo significant morphological, structural, and functional changes, as indicated by the upregulation of uncoupling protein 1 (*UCP1*). This phenomenon, known as “white fat browning”, leads to the formation of beige adipocytes [3,4], which are in an intermediate state between white and brown adipose tissues. Beige and brown fat have smaller adipocyte cells, which facilitate adaptive thermogenesis and increase oxidative energy production [5]. Considering the physiological and functional similarities between pigs and humans, studying the fat metabolism and energy-balance regulatory mechanisms in *Hezuo pigs* under hypothermic conditions may offer new insights and potential therapeutic targets for human diseases such as obesity and metabolic syndrome, which are related to energy-metabolism dysregulation. Research shows that when piglets are kept at a low temperature of 4 °C for 3 days, their white subcutaneous adipose tissue turns beige [6]. The impact of cold exposure on adipose tissue dynamics is complex. It includes promoting lipid metabolism, activating brown adipose tissue, and altering adipocyte morphology. Individual variability, as well as the intensity and duration of cold exposure, significantly influence these effects [7,8,9,10,11,12]. Moreover, fats in different parts of the body perform unique functions, such as thermogenesis and energy storage [13], and the regulatory mechanisms underlying these processes may vary.

In mammals, adipose tissue can be mainly classified into two categories according to its anatomical distribution: subcutaneous adipose tissue (SAT), located beneath the dermal layers, and visceral adipose tissue (VAT), surrounding the internal organs [14]. VAT is characterized by a higher density of inflammatory and immune cells and a greater proportion of larger adipocytes compared to SAT [15]. High temperature promotes the accumulation of VAT in pigs but reduces SAT [16]. However, the differences in the lipid metabolism response of SAT and VAT to low temperature remain unclear.

To better understand the impact of low temperatures on the lipid metabolism of SAT and VAT in *Hezuo pigs*, we selected specific representative locations of SAT and VAT, namely inguinal fat (IF) and perirenal fat (PF). We investigated the histomorphological differences, thermogenic functions, and mRNA transcriptome profiles of IF and PF in *Hezuo pigs* exposed to cold for various durations. This research aims to provide in depth insights into the genetic mechanisms underlying the cold tolerance traits of *Hezuo pigs*, which will contribute to a more comprehensive understanding of the cold adaptation process in these pigs and potentially facilitate the improvement of cold resistance in pig breeding programs.

## 2. Materials and Methods

### 2.1. Animals and Experimental Design

For this investigation, thirty-six, 75 days of age *Hezuo pigs*, exhibiting comparable body weights and deemed clinically healthy, were selected. Following a 7-day acclimation phase under ambient conditions, the animals were weighed and subsequently allocated randomly into two experimental arms: a control cohort and a cold-exposure cohort, each comprising 18 subjects. The low-temperature group was maintained in an environment equipped with a temperature control system set to (−15 ± 2) °C, while the control group was kept at room temperature (23 ± 2) °C. At each time point (12 h, 24 h, 48 h, 5 d, 10 d, and 15 d), three animals per group were euthanized, and their preslaughter weights were recorded. Throughout the experimental period, experimental animals were provided ad libitum access to suckling pig feed (Lanzhou Zhengda Co., Ltd., Lanzhou, China) and lukewarm water.

### 2.2. Sample Collection

IF and PF tissues were collected from *Hezuo pigs*, and a portion of the samples was fixed in a 4% paraformaldehyde solution (Servicebio, Wuhan, China) for hematoxylin-eosin (HE) staining. Another portion was preserved in a 2.5% glutaraldehyde solution (Biosharp, Hefei, China) for subsequent transmission electron microscopy (TEM) analysis (Servicebio, Wuhan, China). The remaining samples were immediately placed in a liquid nitrogen tank for rapid freezing and then stored in a −80 °C freezer for subsequent RNA extraction.

### 2.3. Morphological Observation of IF and PF

IF and PF tissue samples from *Hezuo pigs* at each treatment time point were subjected to HE staining. Adipose tissue was fixed for standard paraffin embedding, and tissue sections with a thickness of approximately 4 μm were prepared. These sections were dewaxed, rehydrated, stained first with hematoxylin and then with eosin, and subsequently dehydrated and mounted for preservation. For each tissue section observed under a 30× magnification, five fields of view were randomly selected, and both the number of adipocytes and their cross-sectional area were quantified and normalized using Image-J software (1.51j8, National Institutes of Health, Bethesda, MD, USA).

For TEM analysis, adipose tissue samples were fixed in 2% glutaraldehyde solution (Servicebio, Wuhan, China). The samples were protected from light and subsequently post-fixed with 1% OsO_4_ in 0.1 M phosphate buffer (pH 7.4) (Servicebio, Wuhan, China) for two hours at room temperature. After fixation, the samples underwent dehydration, infiltration with resin, and embedding procedures. Ultrathin sections were prepared using a diamond knife, mounted on 150-mesh copper grids, and stained first with a 2% uranyl acetate solution (Servicebio, Wuhan, China) in saturated alcohol and then with 2.6% lead citrate solution (Servicebio, Wuhan, China). The sections were then observed under a transmission electron microscope, and images were captured for subsequent analysis.

### 2.4. Real-Time Quantitative PCR (RT-qPCR)

Total RNA was isolated from tissues using TRIzol reagent (Accurate Biology, Changsha, China) according to the manufacturer’s protocol, cDNA synthesis was performed with the Evo M-MLV Reverse Transcription Premix Kit (Accurate Biology, Changsha, China), and mRNA expression levels were quantified using the SYBR^®^ Premix Ex Taq™ II kit (Accurate Biology, Changsha, China). The reaction system consisted of 10 μL of 2× SYBR Green Pro Taq HS Premix, 0.8 μL each of forward and reverse primers, and 1 μL of cDNA template, and ddH_2_O was added to a final volume of 20 μL. The reaction conditions were as follows: initial denaturation at 95 °C for 30 s; 40 cycles of denaturation at 95 °C for 5 s, annealing at 60 °C for 30 s; and melting curve analysis starting with 95 °C for 1 min, 60 °C for 30 s, and finally, 95 °C for 30 s. Each sample was analyzed in triplicate. Each group had three biological replicates, and relative gene expression was calculated using the 2^⁻ΔΔCt^ method [17].

The quantification of mRNA expression levels was determined using GAPDH as the internal control. The primers for the reaction were synthesized by GENEWIZ (Suzhou, China), and the details of primers are presented in Table 1.

### 2.5. mRNA Library Construction and Sequencing

The total RNA was extracted from tissues using TRIzol (TransGen Biotech, Beijing, China) reagent according to the kit instructions. RNA integrity was assessed with the RNA Nano 6000 Assay Kit on the Bioanalyzer 2100 system (Agilent Technologies, Inc., Santa Clara, CA, USA). Briefly, mRNA was purified from total RNA using poly-T oligo-attached magnetic beads. Fragmentation was performed using divalent cations at elevated temperature in 5X First Strand Synthesis Reaction Buffer. First-strand cDNA was synthesized with random hexamer primers and M-MuLV Reverse Transcriptase (RNase H-). Second-strand cDNA synthesis was subsequently performed with DNA Polymerase I and RNase H. Remaining overhangs were converted to blunt ends by exonuclease/polymerase activities. After the adenylation of the 3′ ends of DNA fragments, hairpin-loop-structured adaptors were ligated in preparation for hybridization. To preferentially select cDNA fragments 370–420 bp in length, the library fragments were purified using the AMPure XP system, Indianapolis, IN, USA. Subsequently, PCR was performed using Phusion High-Fidelity DNA Polymerase, Universal PCR Primers, and Index (X) Primer. Finally, PCR products were purified using the AMPure XP system, Indianapolis, IN, USA, and library quality was evaluated on the Agilent Bioanalyzer 2100 system.

After passing the library inspection, the library preparations were sequenced on an Illumina Novaseq platform (Illumina Inc., San Diego, CA, USA), generating 150 bp paired-end reads.

### 2.6. Bioinformatics Analysis for RNA-Seq Data

The low-quality raw sequences were removed; reference genome and gene model annotation files were downloaded directly from a genome database website. Hisat2 (version 2.0.5) was employed to align the cleaned paired-end reads to the swine reference genome (ensembl_110_sus_scrofa_sscrofa11_1_toplevel). Mapped reads from each sample were assembled using StringTie (v1.3.3b) in a reference-based manner. Differential expression analysis between the two groups was performed using the DESeq2 R package (version 1.20.0). Genes identified with padj ≤ 0.05 and |log2FoldChange| ≥ 1 by DESeq2 were categorized.

### 2.7. Functional Enrichment Analysis

Furthermore, a Gene Ontology (GO) enrichment analysis, along with a statistical evaluation of the enrichment in KEGG pathways for differentially expressed genes, was performed using the cluster Profiler R package (3.8.1), utilizing a padj value of less than 0.05 as the criterion for determining significant enrichment.

### 2.8. Screening Differential Genes Involved in Key Lipid Metabolism Pathways

To obtain a more in-depth understanding of the key genes related to lipid metabolism in adipose tissue, we examined the protein–protein interaction (PPI) networks among genes within lipid metabolism-related pathways that were enriched in B_IF and B_PF, utilizing STRING (12.0) and Cytoscape (3.8.0).

### 2.9. Statistical Analysis

Adipocyte number, area, and the results of RT-qPCR were analyzed for statistical significance using SPSS 22.0, and the results were expressed as the mean ± SEM. A *t*-Test was used for the comparisons of gene expression levels between the control group and the cold treatment group, and one-way ANOVA was used for the comparisons of gene expression levels among multiple groups.

## 3. Results

### 3.1. Effect of Low Temperature on the Body Weight of Hezuo Pigs

As shown in Table 2, the low-temperature group had a lower weight-gain rate compared to the control group. No significant differences in body weight were detected between the two groups at 0 h, 12 h, 24 h, and 48 h (*p* > 0.05). However, the low-temperature group had significantly lower body weights at 5 d, 10 d, and 15 d (*p* < 0.05). Furthermore, the average daily body weight gain was significantly lower in the low-temperature group than in the control group (*p* < 0.05).

### 3.2. Effect of Low Temperature on the Morphology in the IF and PF of Hezuo Pigs

HE staining indicated that large-sized adipocytes were observed in the IF and PF of control *Hezuo pigs* (Figure 1A—NIF, NPF). Conversely, exposure to low temperature led to a reduced cross-sectional area and an increased number of adipocytes in the IF and PF of *Hezuo pigs* (Figure 1A—CIF, CPF). Statistical analyses revealed that as the duration of cryopreservation increased, the number of adipocytes increased while their cross-sectional area decreased accordingly. The most significant difference from the control group was observed on day 5. However, with extended exposure to low temperature, a decrease in the number of adipocytes was noted on days 10 and 15, along with a progressive increase in their cross-sectional area (Figure 1B,C).

TEM analyses revealed that both the IF and PF of control *Hezuo pigs* had a single prominent lipid droplet accompanied by a small number of mitochondria (Figure 1D—NIF, NPF). Conversely, compared with the control group, the low-temperature group showed an increase in both the number of lipid droplets and mitochondria (Figure 1E). The most significant difference was observed on day 5, where the adipocytes showed characteristics typical of brown adipocytes (Figure 1D—5d). From day 10 onward, there was a gradual decline in the number of lipid droplets and mitochondria.

### 3.3. Effect of Low Temperature on the UCP3 and PGC-1α mRNA Expression in the IF and PF of Hezuo Pigs

As shown in Figure 2, the mRNA expression levels of thermogenic marker genes *UCP3* and *PGC-1α* in the IF and PF of *Hezuo pigs* treated with low temperature were significantly elevated. The expression levels generally showed an increasing-then-decreasing trend, peaking on the fifth day. Compared with the group at 0 h of low-temperature treatment, the mRNA expression of *UCP3* in IF and *PGC-1α* and *UCP3* in PF was significantly increased at 24 h, 48 h, 5 d, and 10 d (*p* < 0.05), while there were no significant differences at 12 h and 15 d (*p* > 0.05). Similarly, the *PGC-1α* mRNA expression in IF was significantly higher at 24 h, 5 d, and 10 d compared with the group at 0 h of low-temperature treatment (*p* < 0.05), but there were no significant differences at 12 h, 48 h, or 15 d (*p* > 0.05). Notably, there was a transient spike in the mRNA expression of IF *UCP3* and *PGC-1α* at 24 h after cryotreatment.

### 3.4. Quality Control and Characterization of cDNA Library in Hezuo Pigs Adipose Tissue

To further explore the key genes involved in adipose tissue lipid metabolism, adipose tissue samples from *Hezuo pigs* subjected to low-temperature treatment for 24 h, 5 days, and 10 days were collected for RNA sequencing analysis. Information on sequencing data quality control and a summary of RNA-seq alignment are provided in Appendix A, respectively. On average, the cDNA libraries derived from adipose tissue generated between 47,656,243 and 53,062,086 raw reads. Clean read ratio (%) ranged from approximately 95.52% to 97.64% (S1). Q20 scores ranged from 98.71% to 98.89%, and Q30 scores were between 96.45% and 96.98%, with GC content exceeding 48% (S1). Approximately 95% of the reads were successfully mapped to the reference genome, while 91.51% to 93.47% of the reads had a unique genomic location (Appendix A).

### 3.5. Analysis of Differentially Expressed Genes (DEGs)

As shown in Figure 3, compared with the control group, in the groups subjected to cold treatment for 24 h, we identified 473 significantly upregulated and 772 significantly downregulated DEGs in the IF, and ten significantly upregulated and eight significantly downregulated DEGs in the PF. In the groups subjected to cold treatment for 5 d, the comparison revealed 671 significantly upregulated and 967 significantly downregulated DEGs in IF, and 278 significantly upregulated and 601 significantly downregulated DEGs in PF. In the groups subjected to cold treatment for 10 d, there were 51 significantly upregulated and 54 significantly downregulated DEGs in IF, and 51 significantly upregulated and 27 significantly downregulated DEGs in PF.

Data presented in Figure 4 indicate that the quantity of DEGs reached its peak on the 5th day of cold treatment in the B_IF and B_PF groups. A greater number of overlapping genes were observed in the A_IF and B_IF groups, whereas fewer overlapping genes were detected in the other groups.

### 3.6. GO and KEGG Enrichment Analyses

DEGs were classified based on Gene Ontology (GO) terms including biological process (BP), cellular component (CC), and molecular function (MF). Based on the GO and KEGG enrichment analyses, the top 30 most significant GO terms and 20 KEGG pathways were selected and visualized using bar charts and scatter plots. If the number of significant terms and pathways was insufficient, all available ones were visualized (Figure 5 and Figure 6).

Following 24-h low-temperature treatment, the 1245 DEGs in the IF were significantly enriched in 223 BP terms, 26 CC terms, and 35 MF terms. Significantly enriched GO terms were related to extracellular matrix organization, extracellular structural organization, and growth factor interactions. Furthermore, KEGG pathway analysis indicated that the DEGs were notably enriched in key KEGG signaling pathways, including the PI3K–Akt signaling pathway, ECM–receptor interactions, and TNF signaling pathway, among others (Figure 5A,B). The 18 DEGs in the PF showed significant enrichment in three BP terms and two MF terms. Significantly enriched GO terms were associated with fatty acid transport, monocarboxylic acid metabolic processes, and polysaccharide binding. Additionally, KEGG pathway analysis revealed that the DEGs were predominantly enriched in several key signaling pathways, including the AMPK signaling pathway, adipocytokine signaling pathway, FoxO signaling pathway, PI3K–Akt signaling pathway, among others (Figure 6A,B).

After 5-day low-temperature treatment, a total of 1638 DEGs in the IF were notably enriched across 465 BP terms, 18 CC terms, and 25 MF terms. Significantly enriched GO terms were mainly related to extracellular matrix organization, as well as activities related to extracellular matrix and receptor regulation. Furthermore, KEGG pathway analysis showed that DEGs were significantly enriched in KEGG signaling pathways such as the PI3K–Akt signaling pathway, ECM–receptor interactions, PPAR signaling pathway, regulation of lipolysis in adipocytes, and AMPK signaling pathway, among others (Figure 5C,D). In the PF, a total of 879 DEGs were found to be significantly enriched across 373 BP terms, 53 CC terms, and 65 MF terms. Significantly enriched GO terms were mainly associated with processes such as oxidation-reduction, ATP metabolism, and the mitochondrial inner membrane. Furthermore, KEGG pathway analysis revealed critical signaling pathways in which DEGs were significantly enriched, which notably included lipid metabolism pathways like oxidative phosphorylation, the citrate cycle (TCA cycle), and pyruvate metabolism, as well as the three principal pathways associated with neurodegenerative diseases: Parkinson’s disease, Huntington’s disease, and Alzheimer’s disease (Figure 6C,D).

Following 10-day low-temperature treatment, the 105 DEGs in the IF were notably enriched in one BP term. The significantly enriched GO term was related to adipocyte differentiation. Furthermore, KEGG pathway analysis indicated that the DEGs were significantly concentrated in pathways related to TNF signaling pathway, cholesterol metabolism, glycerolipid metabolism, and others (Figure 5E,F). Following 10-day low-temperature treatment, the 78 DEGs in the PF were significantly enriched in four BP terms. Significantly enriched GO terms were related to the regulation of hormone levels and the regulation of wound healing. KEGG pathway analysis showed that DEGs were significantly enriched in KEGG signaling pathways such as the FoxO signaling pathway, the PI3K–Akt signaling pathway, and the PPAR signaling pathway, among others (Figure 6E,F).

### 3.7. Differential Transcriptomic Profiles Between IF and PF

Based on the indicators of morphological changes in adipose tissue, heat production capacity and the number of DEGs, B_IF and B_PF were selected to study the differences lipid metabolism levels between IF and PF under low-temperature conditions. As illustrated in Figure 7A, a total of 359 genes were identified as being shared between the IF and PF samples. Additionally, 1279 DEGs were exclusively observed in the IF samples, while 520 DEGs were exclusively identified in the PF samples.

The GO analyses revealed that the genes that were found to be common to both the IF and PF were mainly enriched in the processes of chromosome segregation and the regulation of the cell cycle (Figure 7B). The DEGs in the IF tissues were predominantly enriched in extracellular matrix organization, axon development, and cell adhesion molecule binding (Figure 7D). The DEGs in the PF tissues were predominantly enriched in the oxidation-reduction process, mitochondrial inner membrane, and cofactor binding (Figure 7F).

The KEGG analysis revealed that the genes exhibiting overlap between the IF and PF groups demonstrated significant enrichment in pathways related to lipid metabolism such as cell cycle, pyruvate metabolism and cholesterol metabolism (Figure 7C). The DEGs in the IF tissues exhibited significant enrichment in the PPAR signaling pathway, PI3K–Akt signaling pathway, regulation of lipolysis in adipocytes, cAMP signaling pathway, AMPK signaling pathway, and Wnt signaling pathway (Figure 7E). The DEGs in the PF tissues exhibited significant enrichment in oxidative phosphorylation, Parkinson’s disease, Huntington’s disease, and Alzheimer’s disease (Figure 7G).

### 3.8. Screening for Lipid Metabolism-Related DEGs

The specific differentially expressed genes in B_IF and B_PF are shown in Figure 8 and Figure 9. The top 20 DEGs are listed in Table 3.

### 3.9. Validation of RNA-Seq Results by RT-qPCR

To verify the RNA-seq results, ten DEGs, namely TNS4, CHD5, CLPSL2, CDH9, DSG3, LIPG, NRIP3, SAL1, SCD, and KCNK3, were randomly chosen for RT-qPCR analysis. The results showed that the results of RT-qPCR analysis were consistent with those of RNA-seq analysis (Figure 10). This indicated that the gene expression levels determined by RNA-seq were reliable.

## 4. Discussion

Hypothermia is a critical environmental stressor that significantly impacts animal physiological functions and overall production efficiency. Mammals primarily regulate their body temperature in low-temperature environments through two mechanisms: shivering and non-shivering thermogenesis [18]. Adipose tissue, particularly brown fat, plays a central role in non-shivering thermogenesis, contributing approximately 70% of the total heat produced through this process. Studies have demonstrated that cold exposure induces the browning of white adipose tissue (WAT) in mice, characterized by an upregulated expression of the thermogenic marker *UCP1* [19]. Specifically, cold exposure has been shown to promote the browning of inguinal WAT (iWAT) [20]. Experimental evidence reveals that when mice are exposed to 8 °C for one week, noticeable beige coloration appears in their subcutaneous white fat, indicating browning [21]. Similar to classical brown adipocytes, beige adipocytes exhibit multilocular lipid droplets and densely packed mitochondria, expressing *UCP1* and other genes essential for energy expenditure [22,23]. Our research findings demonstrate that cryogenic exposure in *Hezuo pigs* results in reduced body weight gain, decreased adipocyte cross-sectional area, and increased adipocyte number and mitochondrial density in both inguinal fat (IF) and perirenal fat (PF) depots. These morphological and cellular changes suggest that *Hezuo pigs* exhibit a decreased growth rate under cold stress conditions. Moreover, the observed alterations in IF and PF depots indicate that low temperatures may induce browning in these adipose tissues, thereby enhancing the thermogenic capacity of *Hezuo pigs*.

*UCP1* and *PGC-1α* play crucial roles in regulating thermogenesis during white adipose tissue browning, with their expression levels significantly upregulated during this process. Although pigs lack brown adipose tissue and functional *UCP1*, they have developed alternative non-shivering thermogenesis mechanisms to adapt to cold environments [24]. Previous studies have demonstrated that cold adaptation in Tibetan pigs and Min pigs is associated with white adipose tissue browning and *UCP3* upregulation, suggesting the thermogenic potential of *UCP3* in porcine adipose tissue [25]. In the present study, we observed a significant upregulation of *UCP3* and *PGC-1α* mRNA expression in both IF and PF depots of cold-exposed *Hezuo pigs*. This elevated gene expression indicates enhanced thermogenic capacity, which peaked on the fifth day of cold exposure.

Transcriptome analysis revealed a greater number of DEGs in the IF of the hypothermia group compared to PF. The IF depot demonstrated a substantial number of DEGs at both 24 h and 5 days post-exposure, whereas the PF depot showed fewer DEGs at 24 h, with a significant increase observed only at the 5-day time point. These findings suggest that SAT exhibits a more rapid and dynamic response to cold exposure than VAT, potentially defense against hypothermia through diverse molecular adaptation mechanisms. In contrast, VAT may respond slowly, potentially playing a crucial role in long-term cold adaptation. These depot-specific response patterns are consistent with previous studies demonstrating heterogeneous physiological responses to cold exposure across different adipose depots in mice [26,27].

GO and KEGG analyses were conducted to explore the function of the differential genes between groups. The results of the GO analysis indicated that the GO entries related to the differential genes were mainly associated with processes such as cytoplasmic matrix, growth factor binding, adipocyte differentiation, fatty acid transport, and ATP metabolism. The KEGG analysis revealed that these genes were predominantly enriched in classical lipid metabolism pathways, including the AMPK signaling pathway, PI3K–Akt signaling pathway, ECM–receptor interaction, FoxO signaling pathway, and PPAR signaling pathway. Lipid metabolism encompasses various pathways, with lipid synthesis and catabolism as core components. Different signaling pathways play distinct roles in regulating lipid metabolism. For instance, the activation of the AMPK signaling pathway is known to enhance energy expenditure, simultaneously reducing lipid accumulation and adipogenesis [28,29]. The PI3K–Akt signaling pathway, which is widely present in diverse cell types, plays a crucial role in governing cellular glucose and lipid metabolism. It specifically regulates the activation of key enzymes involved in these metabolic pathways [30,31,32,33]. On the other hand, the ECM–receptor interaction pathway effectively modulates adipocyte development and function. It does so by influencing intracellular signaling pathways, shaping the architecture of adipose tissue, and regulating the dynamics of lipid deposition [34,35]. The SIRT1-AMPK-FOXO1 signaling cascade regulates the expression of lipid metabolism-related genes (such as *PPARγ*, FASN, HSL, and LPL), thereby inhibiting intramuscular fat deposition in bovine muscle [36]. Additionally, FoxO1 has been demonstrated to suppress the transcription of *SREBP1*, leading to a downregulation of the mRNA expression of genes essential for de novo fatty acid synthesis (including *FASN*, *ACC*, *SCD1*, etc.), ultimately limiting lipid synthesis [37]. Moreover, the PPAR pathway is closely intertwined with the regulation of lipid metabolism [38,39]. In this study, the results suggest that in *Hezuo pigs* exposed to low temperatures, the expression of lipid metabolism-related genes is modulated via the activation of several key signaling pathways, namely AMPK, PI3K–Akt, ECM–receptor interaction, FoxO, and PPAR. These pathways collectively act to suppress fat synthesis and promote lipolysis, thus providing the necessary energy to meet the metabolic demands under low-temperature conditions.

In this study, we aimed to explore the molecular mechanisms underlying the response to cold conditions in *Hezuo pigs*. We analyzed the DEGs in IF and PF. Our findings revealed that the co-expressed DEGs in both IF and PF were significantly enriched in terms related to cell differentiation. Additionally, they were associated with pertinent pathways involved in lipid metabolism and the cell cycle. Specifically, the IF-specific DEGs were prominently concentrated in several key lipid metabolism pathways. On the other hand, genes that were uniquely expressed in PF were significantly enriched in pathways associated with well-known neurodegenerative diseases, such as Huntington’s disease, Parkinson’s disease, and Alzheimer’s disease. We also identified several key genes that are potentially involved in the mechanism by which adipose tissue responds to hypothermia. In terms of lipid metabolism regulation, *FABP4* plays a crucial role in maintaining adipocyte homeostasis. It regulates lipolysis and lipogenesis through interactions with HSL and *PPARγ* [40]. *WNT10B* serves as a pivotal molecular switch, inhibiting adipocyte differentiation. Intriguingly, when its expression is knocked down, it promotes the expression of *UCP1* in mouse brown adipose tissue [41]. Regarding energy metabolism, phosphoenolpyruvate carboxykinase, encoded by the *PCK1* gene, acts as the rate-limiting enzyme for gluconeogenesis [42]. Previous research has shown that appropriate cold stimulation can substantially enhance the expression of *PCK1* in the liver of broilers [43].

Furthermore, other genes such as *PLIN1* can activate PPARs. *LEPR* is widely distributed throughout the body and serves as a key mediator of leptin’s action. Both *LEP* and *ADIPOQ* are factors secreted by adipocytes and are closely associated with body fat content. Collectively, the above results suggest that under cold conditions, the lipid metabolism within the IF of *Hezuo pigs* is significantly enhanced. This increased lipid metabolism provides the necessary energy for the pigs to adapt to the cold environment. Conversely, in the PF of *Hezuo pigs*, the hub genes *ATP5F1A*, *ATP5PO*, *SDHB*, *NDUFS8*, *SDHA*, and *COX5A* are co-expressed. These genes are important in the context of the three major neurodegenerative disorders and are downregulated following cold treatment. Previous research has demonstrated that adipose tissue can influence brain development, cognitive function, and the risk of neurodegenerative diseases throughout an organism’s lifespan [44]. In this context, the metabolic changes in iWAT induced by cold exposure may have a positive impact on neurodegenerative diseases [45]. In conclusion, the functions of these candidate genes related to the lipid metabolism of *Hezuo pigs* remain to be further explored. Such exploration is essential for uncovering the genetic mechanisms that underlie the cold-resistance traits of local pig breeds.

## 5. Conclusions

Hypothermia diminished the cross-sectional area of adipose cells while simultaneously increasing the number of IF and PF adipocytes, which elevated both the quantity of lipid droplets and mitochondria, ultimately sustaining the body heat in *Hezuo pigs*. On day five of hypothermic exposure, *Hezuo pigs* displayed the most significant morphological changes in adipose tissue, concurrent with maximal thermogenesis and elevated lipid metabolism. IF regulates lipid metabolism primarily via genes including *FABP4*, *WNT10B*, *PCK1*, *PLIN1*, *LEPR*, and *ADIPOQ*, providing energy for cold adaptation, whereas PF positively influences in vivo degenerative diseases mainly through genes such as *ATP5F1A*, *ATP5PO*, *SDHB*, *NDUFS8*, *SDHA*, and *COX5A*.

## Figures and Tables

**Figure 1 cells-14-00392-f001:**
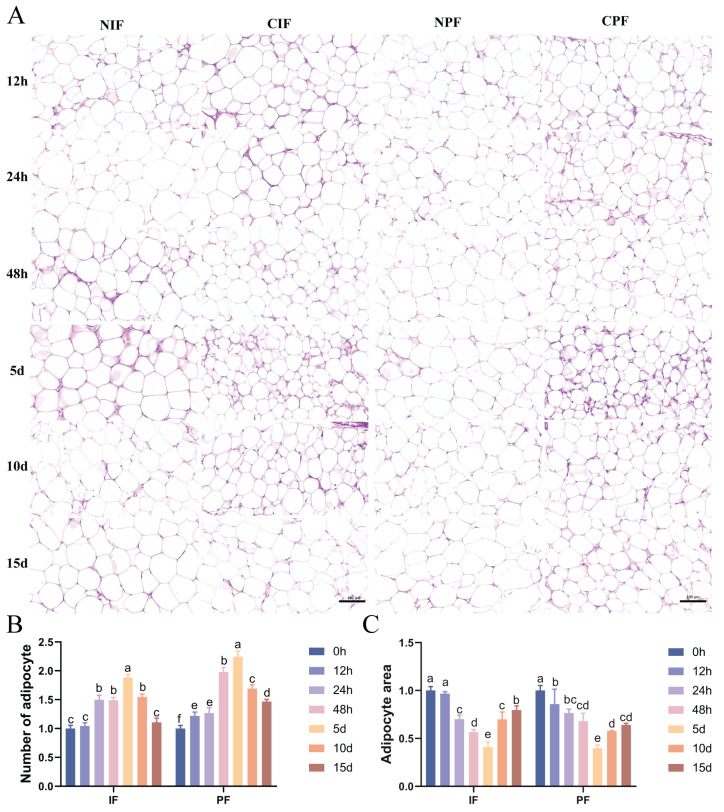
Effect of low temperature on the morphology of IF and PF in *Hezuo pigs*. (**A**) HE staining of adipose tissue, scale bar 100 μm; (**B**) number of adipocytes; (**C**) adipocytes cross-sectional area; (**D**) TEM of adipose tissue (LD: lipid droplet; arrows: mitochondria; and (**E**) number of lipid droplets. NIF: normal inguinal fat; CIF: cold inguinal fat; NPF: normal perirenal fat; CPF: cold perirenal fat. The same letter indicates a non-significant difference (*p* > 0.05) and different letters indicate a significant difference (*p* < 0.05).

**Figure 2 cells-14-00392-f002:**
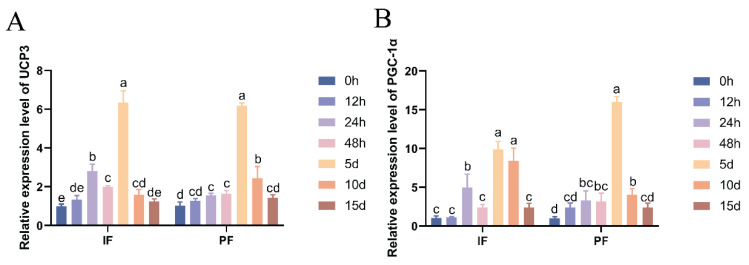
Effect of low temperature on the *UCP*3 (**A**) and *PGC*-1α mRNA (**B**) expression in the IF and PF of *Hezuo pigs*. The same letter indicates a non-significant difference (*p* > 0.05) and different letters indicate a significant difference (*p* < 0.05).

**Figure 3 cells-14-00392-f003:**
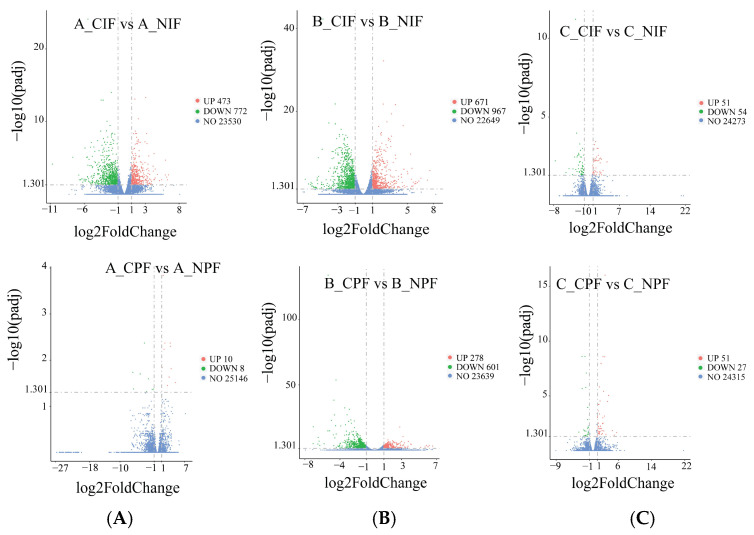
The volcano blot of DEGs in IF and PF of *Hezuo pigs* at 24 h, 5 d, and 10 d of cold treatment. (**A**) Low-temperature treatment for 24 h; (**B**) low-temperature treatment for 5 d; and (**C**) low-temperature treatment for 10 d.

**Figure 4 cells-14-00392-f004:**
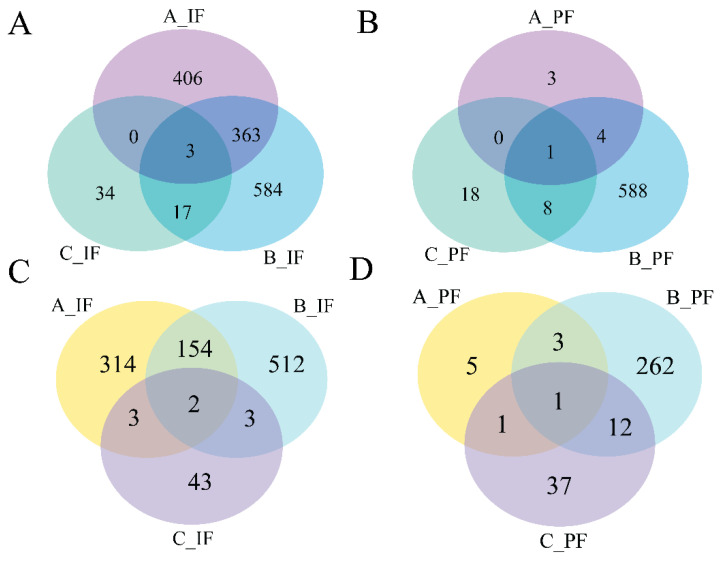
Venn diagrams of the number of DEGs between groups: (**A**) downregulated genes of IF; (**B**) downregulated genes of PF; (**C**) upregulated genes of IF; and (**D**) upregulated genes of PF.

**Figure 5 cells-14-00392-f005:**
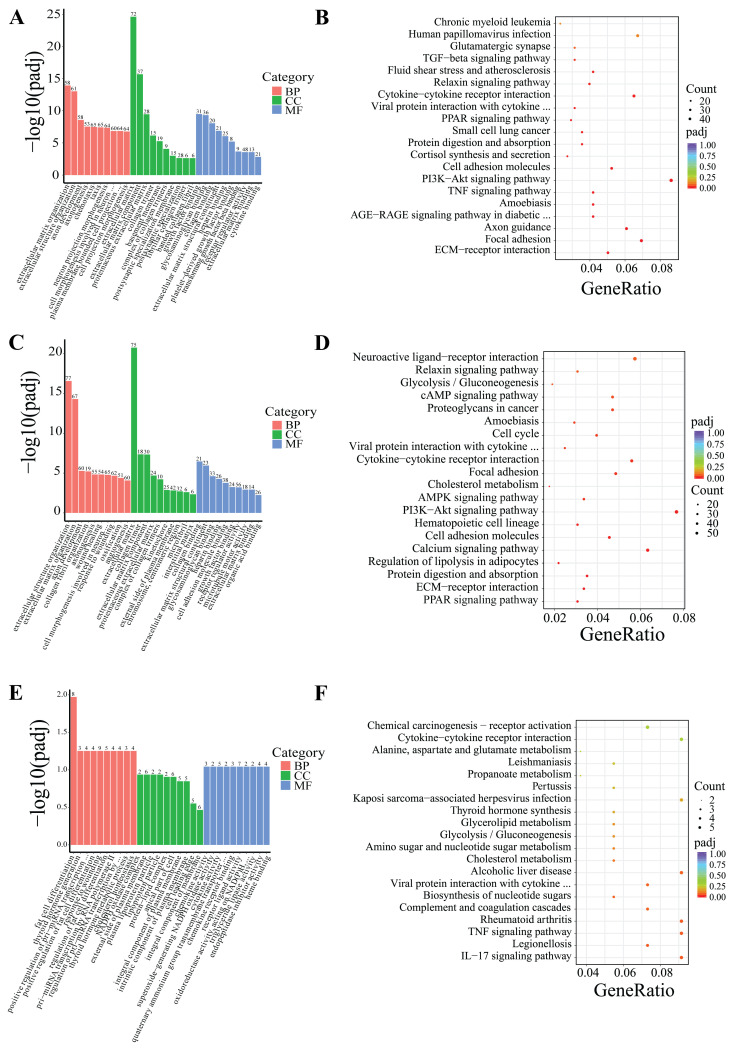
GO and KEGG enrichment analysis of DEGs in IF. (**A**,**B**) Low-temperature treatment for 24 h; (**C**,**D**) low-temperature treatment for 5 d; and (**E**,**F**) low-temperature treatment for 10 d.

**Figure 6 cells-14-00392-f006:**
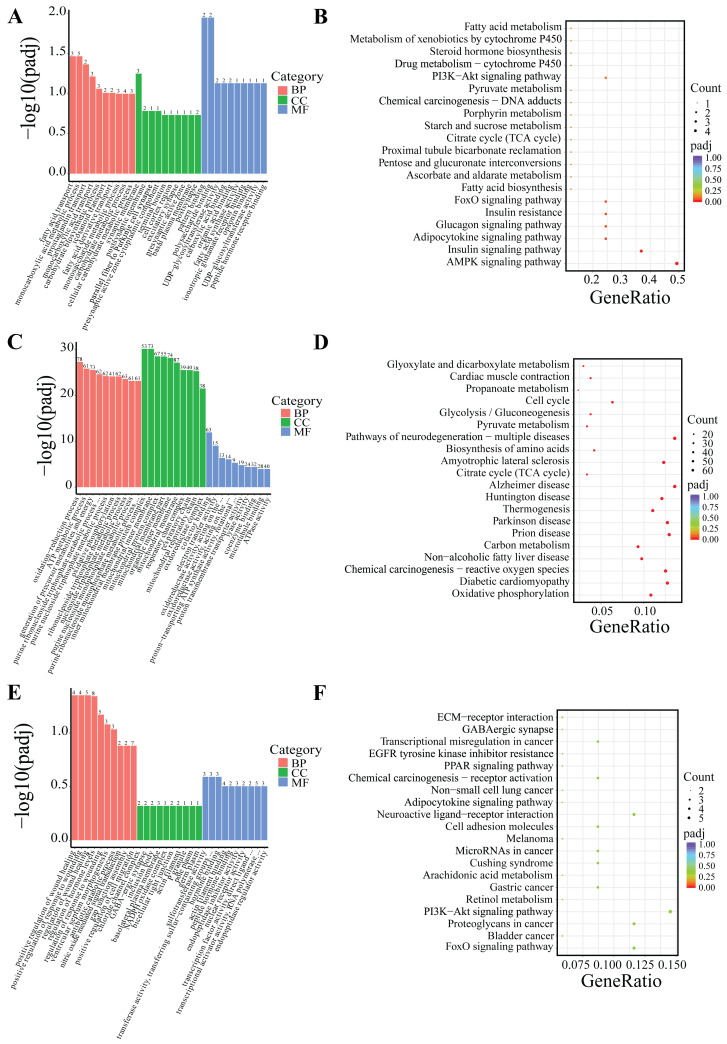
GO and KEGG enrichment analysis of DEGs in PF. (**A**,**B**) Low-temperature treatment for 24 h; (**C**,**D**) low-temperature treatment for 5 d; and (**E**,**F**) low-temperature treatment for 10 d.

**Figure 7 cells-14-00392-f007:**
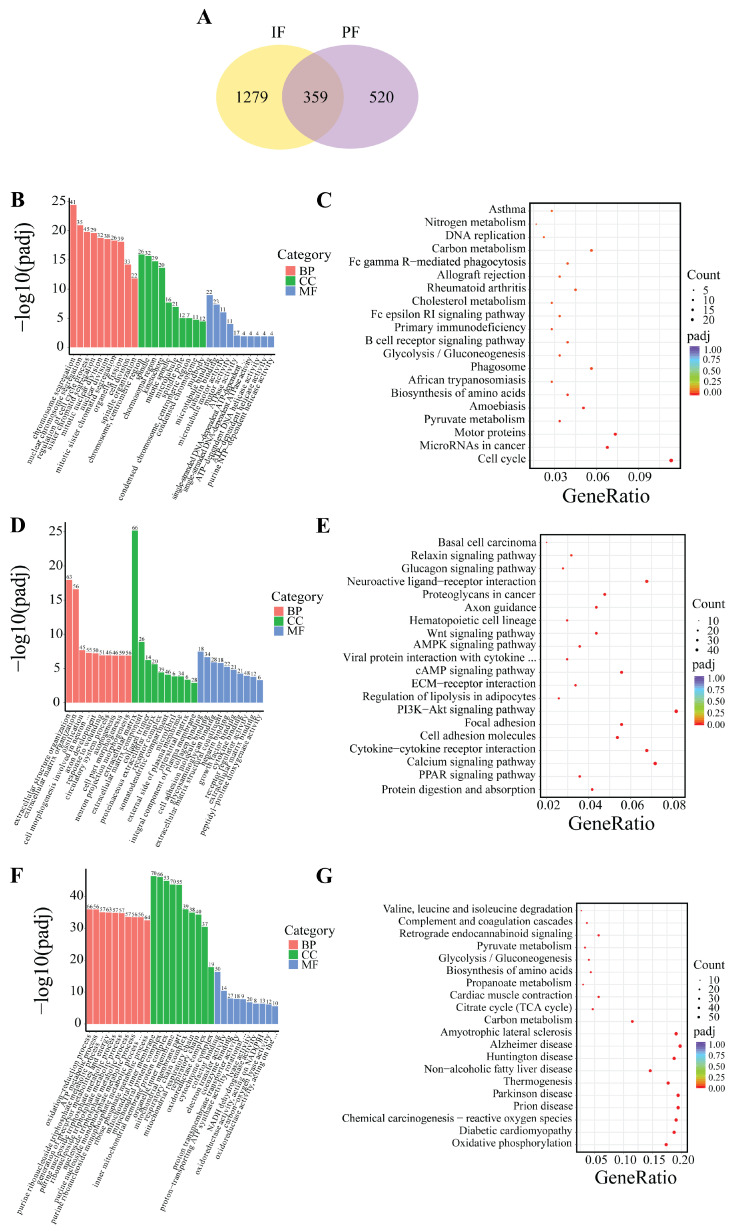
DEG Venn and enrichment analysis between B_IF and B_PF. (**A**) DEG Wayne plots; (**B**,**C**) GO and KEGG enrichment analysis of genes co-expressed in B_IF and B_PF; (**D**,**E**) GO and KEGG enrichment analysis of genes specifically expressed in B_IF; and (**F**,**G**) GO and KEGG enrichment analysis of genes specifically expressed in B_PF.

**Figure 8 cells-14-00392-f008:**
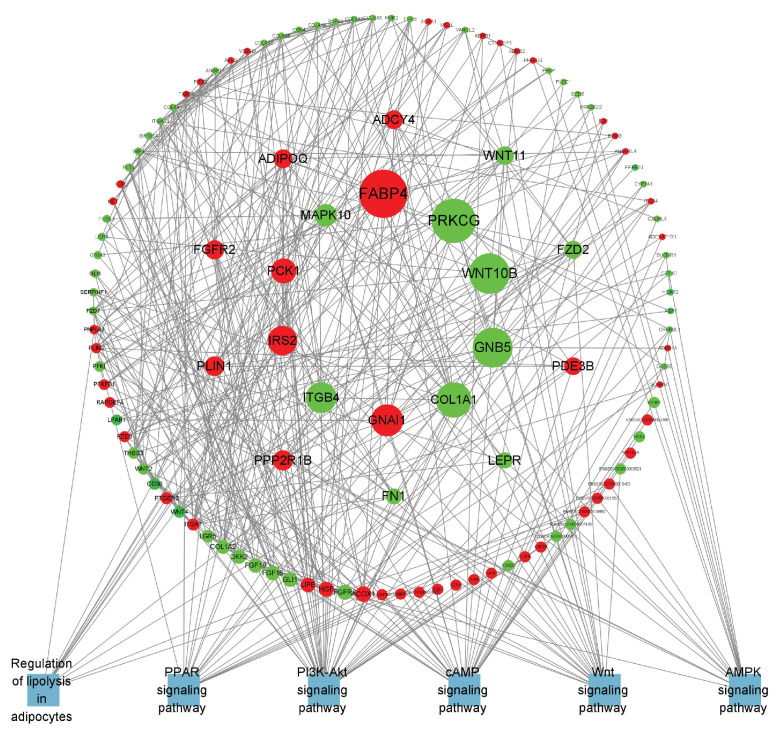
PPI network of B_IF-specific expressed genes. Red represents upregulated genes, green represents downregulated genes, blue represents the pathway where the gene is located, and the larger the range of the node, the larger the Betweenness value of the node.

**Figure 9 cells-14-00392-f009:**
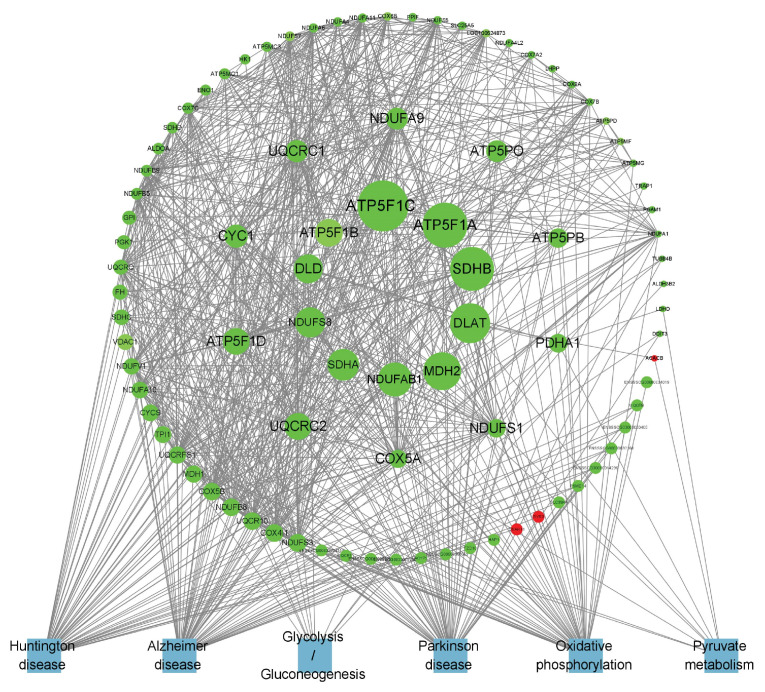
PPI network of B_PF-specific expressed genes.

**Figure 10 cells-14-00392-f010:**
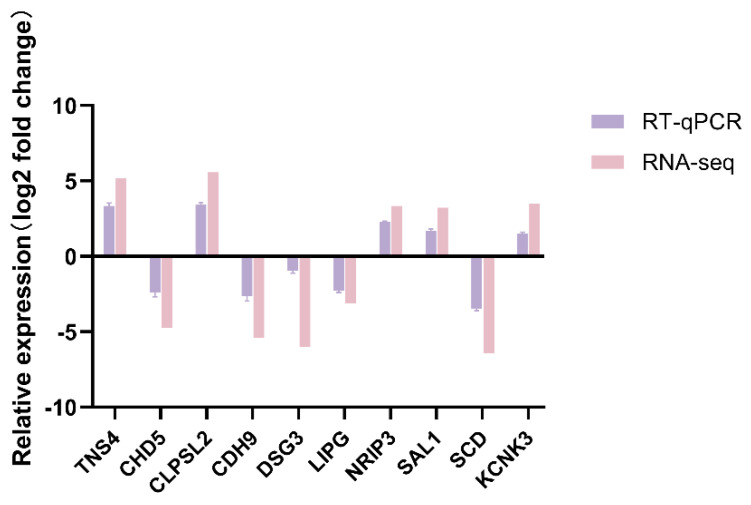
RT-qPCR verification of DEGs.

**Table 1 cells-14-00392-t001:** A list of the primers used in RT-qPCR.

Gene Name	Primer Sequences (5′-3′)	Length (bp)	Transcript No.
*UCP*3	F: TCACCTTCAGGACACGTTCG R: AGGCATCCATCCTAGTGGGT	84	NM_214049.1
*PGC*-1α	F: CATGTGCAACCAGGACTCTGT R: GCGTCTCTGTGAGAACTGCTA	269	NM_213963.2
*TNS*4	F: GCAGCCCACCATGAAGTTTG R: TGTCCCTCACGATGAAAGCC	122	XM_003131470.4
*CHD*5	F: CTCCAGATGTCCGAGCGAAG R: GGGCACCTACCGGTCACATA	194	XM_021095234.1
*CLPSL*2	F: ATCAACTTGGACCTCGGTGG R: CCTTGGCAGGGACAGACAAT	107	XM_005665918.3
*CDH*9	F: ACGATTACCTCAGCGACTGG R: GAAAGCAGGCCACTCCATCT	150	XM_021076628.1
*DSG*3	F: AAGCAGACACACGGGAGAAG R: CACCACTCACAACCAGACGA	86	XM_003356391.4
*LIPG*	F: GAAACCTAGTGGGATGGGGAT R: GCCAAGAGTGCCTTTCATCAC	82	NM_001243029.1
*NRIP*3	F: GATCCCTTTTGTGGAGACGGT R: TCTGTCAACCTGGTGTGTGT	105	XM_005667042.3
*SAL*1	F: CCTCTTCCCACAAGGAAGCA R: CCAAGACACGGATATGCTCCA	162	NM_213814.2
*SCD*	F: AAATCTTCTGGGAAAGCCCCTG R: TCCCCTGACAAGTTACCTGC	111	NM_213781.1
*KCNK*3	F: TCTACTTCGCCATCACCGTC R: ATGCACGAGAAGAAGCCGAT	253	NM_001315680.1
*GAPDH*	F: AGTATGATTCCACCCACGGC R: TACGTAGCACCAGCATCACC	139	NM_001206359.1

**Table 2 cells-14-00392-t002:** Effect of low temperature on the body weight of *Hezuo pigs*.

Term	Control Group (kg)	Low-Temperature Group (kg)	*p*-Value
0 h	5.17 ± 0.03	5.18 ± 0.04	0.718
12 h	5.18 ± 0.04	5.18 ± 0.04	0.922
24 h	5.24 ± 0.06	5.21 ± 0.05	0.495
48 h	5.44 ± 0.09	5.27 ± 0.07	0.057
5 d	5.98 ± 0.14	5.54 ± 0.08	0.008
10 d	6.58 ± 0.18	5.78 ± 0.27	0.013
15 d	7.70 ± 0.20	6.24 ± 0.28	0.002
Average daily weight gain (g/d)	169.00 ± 11.36	70.33 ± 16.74	0.001

**Table 3 cells-14-00392-t003:** Betweenness values for the first 20 DEGs in B_IF and B_PF.

Groups	Genes	Betweenness	Groups	Genes	Betweenness
B_IF	*FABP4*	1872.3813	B_PF	*ATP5F1C*	153.1263
	*PRKCG*	1696.4803		*ATP5F1A*	132.4847
	*WNT10B*	1520.4729		*SDHB*	128.6541
	*GNB5*	1493.9647		*DLAT*	115.2456
	*COL1A1*	1285.897		*MDH2*	107.9385
	*GNAI1*	1145.1943		*NDUFAB1*	94.6571
	*ITGB4*	1084.635		*SDHA*	86.3341
	*IRS2*	1029.1504		*NDUFS8*	81.3528
	*PCK1*	827.73615		*DLD*	77.0280
	*MAPK10*	700.68866		*ATP5F1B*	73.2094
	*PPP2R1B*	608.68066		*UQCRC2*	70.0166
	*PLIN1*	585.4062		*ATP5F1D*	69.3359
	*FGFR2*	564.2501		*CYC1*	57.2409
	*ADIPOQ*	546.2029		*UQCRC1*	53.8775
	*ADCY4*	540.1115		*NDUFA9*	52.7184
	*WNT11*	530.37274		*ATP5PO*	52.2509
	*FZD2*	510.57275		*ATP5PB*	44.4790
	*PDE3B*	501.63425		*PDHA1*	41.9882
	*LEPR*	422.55783		*NDUFS1*	41.1655
	*FN1*	412.8051		*COX5A*	39.9688

## Data Availability

The data presented in this study are available on request from the corresponding author before 1 May 2025. The dataset supporting the conclusions of this article is available in the NCBI Sequence Read Archive. URL (accessed on 1 May 2025) (SRA accession number: PRJNA1194207, https://www.ncbi.nlm.nih.gov/sra/PRJNA1194207).

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
