# Peer review of "Adaptive Thermogenesis and Lipid Metabolism Modulation in Inguinal and Perirenal Adipose Tissues of Hezuo Pigs in Response to Low-Temperature Exposure"

_cells, 2025, doi:10.3390/cells14060392_

Round 1
Reviewer 1 Report
Comments and Suggestions for Authors
General comments:
This study evaluated the effect of low environmental temperatures (-15 ºC vs. 23 ºC, group control, at different duration length) on histomorphological variances, thermogenic function, and mRNA transcriptome profiles of inguinal and perirenal fat from (75-day olds) Hezuo pigs. This study highlights the timeline effect on these outcomes ensuring the novelty of the findings. Overall, the manuscript is well presented regarding all sections. The introduction contextualizes the subject and objective of the study. The study design is well described in M&M, including the data about quality control reported in the supplementary file. Ten elucidative figures support the findings. The discussion is adapted to the results, succinctly, without redundancy and supported by adequate literature. This can help the reader to extract the core information for further research. The conclusions are supported by the results.
Specific comments:
Please confirm that all abbreviations and their definitions were correctly applied.
Fig. 1B and C: Please add the p-value for different letters (a-d) in the caption.
Fig. 2A and B: Please add the p-value for different letters (a-d) in the caption.
Fig. 5, 6 and 7: Please improve the readability of the x-axis of these figures (probably due to pdf conversion).
Author Response
Comments 1: Please confirm that all abbreviations and their definitions were correctly applied.
Response 1: We have checked and confirmed that all abbreviations used in the manuscript and their definitions are are accurate and correctly applied.
Comments 2: Fig. 1B and C: Please add the p-value for different letters (a-d) in the caption.
Response 2: We have added the p-value for different letters in Fig. 1B and C of the revised manuscript (Row 211-212).
Comments 3: Fig. 2A and B: Please add the p-value for different letters (a-d) in the caption.
Response 3: We have added the p-value for different letters in Fig. 2A and B of the revised manuscript (Row 234-235).
Comments 4: Fig. 5, 6 and 7: Please improve the readability of the x-axis of these figures (probably due to pdf conversion).
Response 4: According to the comment, we have improve the readability of the x-axis of Figures 5, 6 and 7 to ensure that the content is clear and easy to read (Row 312, 316 and 339).
Reviewer 2 Report
Comments and Suggestions for Authors
An article entitled to “Adaptive Thermogenesis and Lipid Metabolism Modulation in Inguinal and Perirenal Adipose Tissues of Hezuo Pigs in Response to Low-Temperature Exposure” delineated that the effect of cold exposure on histomorphological variances, thermogenic function, and mRNA transcriptome profiles of inguinal fat (IF) and perirenal fat (PF) in Hezuo pigs The following comments should be addressed.
1. Why is it important to understand the responses and/or changes of histomorphological differences, thermogenic function, and mRNA transcriptome profiles to cold exposure in Hezuno pig? How did the changes caused by cold exposure in Hezuno pigs influece Hezuno pigs or humans? What changes did the effects of low temperature, such as the SIRT1-AMPK-FOXO1 signaling cascade, found through GO and KEGG analyzes ultimately cause in lipid metabolism in Hezuno pigs?
2. What is your conclusion? Please tell us your conclusion about low temperature exposure time.
3. There was no information about research ethics. Please add the Institutional Review Board Statement and approval number.
4. Changes of body weight and food intake during cold exposure need to be shown.
5. Even at the same adipose depot, adipocyte number, adipocyte area, and mRNA expression were different depending on the exposure to low temperature. For example, 12 h – 10 d cold exposure significantly increased adipocyte number in IF but not 15 d compared to 0 h. Therefore, the statement made in the abstract, “Adipocyte cross-sectional area in IF and PF was found to be decreased, whereas adipocyte number was increased in the low-temperature cohort” is incorrect. Please check out and correct your statements regarding adipocyte number, adipocyte area, and mRNA expression.
6. There was no scale bar in H&E staining in adipose depots, Figure 1.
7. Cold exposure-induced mitochondria abundancy was very difficult to detect. Please express the number and size of mitochondria graphically as adipocyte number or adipocyte area.
Author Response
Comments 1. Why is it important to understand the responses and/or changes of histomorphological differences, thermogenic function, and mRNA transcriptome profiles to cold exposure in Hezuno pig? How did the changes caused by cold exposure in Hezuno pigs influece Hezuno pigs or humans? What changes did the effects of low temperature, such as the SIRT1-AMPK-FOXO1 signaling cascade, found through GO and KEGG analyzes ultimately cause in lipid metabolism in Hezuno pigs?
Response 1: Thank you for your important comments.
1) In cold environments, animals typically employ increased thermogenesis to maintain core body temperature. Non-shivering thermogenesis (NST), primarily driven by lipid metabolism, represents a crucial adaptive mechanism for prolonged cold exposure. This has made NST a focal point for understanding cold resistance in animals. Mammalian adipose tissue comprises of white, brown, and beige fat depots. White adipocytes are large and primarily store energy, whereas brown and beige adipocytes are smaller and directly contribute to heat production. Upon cold exposure, white adipocytes undergo a browning process, differentiating into beige adipocytes. This transformation enhances thermogenesis, thereby maintaining core body temperature and providing cold protection. Investigating the morphological variations, thermogenic capacity, and mRNA transcriptome profiles of adipose tissue in Hezuo pigs can provide insights into the genetic mechanisms governing cold adaptation. These insights have been integrated into the introduction of revised manuscript (Row 47-50).
2) The physiological and functional similarities between Hezuo pigs and humans render them a valuable model for investigating cold-induced physiological adaptations. Specifically, the study of Hezuo pigs under low-temperature conditions offers insights into potential human responses to cold exposure. For instance, the alterations in lipid metabolism and energy balance regulation observed in Hezuo pigs may provide novel therapeutic targets for obesity, metabolic syndrome, and other energy metabolism-related disorders in humans. In essence, the low-temperature-induced changes in Hezuo pigs have significant implications for their growth, health, and physiological functions, as well as for understanding human physiological adaptation and disease prevention in cold environments. This information has been incorporated into the revised manuscript (Row 59-64).
3) Upon cold exposure in Hezuo pigs, multiple lipid metabolism signaling pathways are upregulated, leading to the suppression of fat deposition and the promotion of lipolysis. This is achieved through the regulation of key lipid metabolism-related genes, including LEP, PPARγ, FAS, HSL, and LPL, thereby providing the necessary energy for thermoregulation. Data on the effects of low temperature on the expression of lipid metabolism genes previously studied by our research group are presented below:
As shown in Figure 1, compared with the control group, LEP mRNA expression was highly significantly reduced (P<0.01), AMPK mRNA expression was highly significantly elevated (P<0.01), and PPARγ mRNA expression was significantly reduced (P<0.05) in AF of low-temperature Hezuo pigs; in IF, LEP mRNA expression was highly significantly reduced (P<0.01), and AMPK mRNA expression was significantly higher (P<0.05), and PPARγ mRNA expression was significantly lower (P<0.01); the expression of ADPN and FAT mRNA in the IF of low-temperature Hezuo pigs was extremely significantly higher (P<0.01), and the difference in AF was not significant (P>0.05).
As shown in Figure 2, LPL activity in AF and IF of low-temperature Hezuo pigs was highly significantly elevated compared with the control group (P<0.01), and FAS activity was highly significantly lower than that of the control group (P<0.01); HSL activity in IF was highly significantly elevated compared with that of the control group (P<0.01), and HSL activity in AF showed a tendency to be higher compared with that of the control group, but the difference was not significant (P>0.05).
Currently being prepared for publication. Revisions to the manuscript have been made to incorporate these findings (Row 416-421).
Comments 2. What is your conclusion? Please tell us your conclusion about low temperature exposure time.
Response 2: Thank you for pointing this out. The findings of our investigation indicate that lipid metabolism exhibited peak activity within the adipose tissue of Hezuo pigs on the fifth day following exposure to hypothermia. This information has been incorporated into the revised manuscript (Row 451-454).
Comments 3. There was no information about research ethics. Please add the Institutional Review Board Statement and approval number.
Response 3: We are deeply grateful for your important comment. This information has been incorporated into the revised manuscript (Row 467-469).
Comments 4. Changes of body weight and food intake during cold exposure need to be shown.
Response 4: Thanks for your insightful comments. Body weight and food intake are likely key factors influencing the energy metabolism of animals in response to cold stress. We have supplemented the change of body weight of Hezuo pigs during hypothermic exposure (Row 181-188). However, we acknowledge a limitation in our experimental design; specifically, all pigs were fed ad libitum, precluding the quantification of individual feed intake.
Comments 5. Even at the same adipose depot, adipocyte number, adipocyte area, and mRNA expression were different depending on the exposure to low temperature. For example, 12 h – 10 d cold exposure significantly increased adipocyte number in IF but not 15 d compared to 0 h. Therefore, the statement made in the abstract, “Adipocyte cross-sectional area in IF and PF was found to be decreased, whereas adipocyte number was increased in the low-temperature cohort” is incorrect. Please check out and correct your statements regarding adipocyte number, adipocyte area, and mRNA expression.
Response 5: We appreciate your comments. Our adipocyte quantification revealed that all cryoexposure durations (12 h to 15 d) induced an increase in adipocyte number. However, the 12 h and 15 d time points exhibited a less pronounced and statistically insignificant increase, and were thus designated as 'c' (Row 17-24).
Following your recommendations, we have revised the presentation of adipocyte number, adipocyte area, and mRNA expression data in the updated manuscript.
Comments 6. There was no scale bar in H&E staining in adipose depots, Figure 1.
Response 6: Thank you for your important comments. In our mapping strategy, we addressed the potential for scale bar redundancy to impede image interpretation. Consequently, a uniform multiplier was applied across all HE-stained maps, and scale bars were positioned within the bottom-right images of the IF and PF tissues. (Row 206, 209).
Comments 7. Cold exposure-induced mitochondria abundancy was very difficult to detect. Please express the number and size of mitochondria graphically as adipocyte number or adipocyte area.
Response 7: We appreciate your suggestion to utilize adipocyte number or area as a proxy for mitochondrial content and size. In our preliminary investigation, we opted for direct visualization of mitochondrial alterations.
Following your recommendation, we have incorporated indirect evidence of mitochondrial abundance through adipocyte quantification, with the corresponding data presented in the revised manuscript (Figures 1-2F).
Round 2
Reviewer 2 Report
Comments and Suggestions for Authors
It seems that authors comment all questions and corrected according to reviewer's opinion.
Comments on the Quality of English Language
Good
Author Response
Comment: The English language can be improved to express the research more clearly.
Response: Thank you for your careful review of the manuscript, and we have improved the language quality of the manuscript to express the research more clearly.